# The Value of ^18^F-FDG-PET-CT Imaging in Treatment Evaluation of Colorectal Liver Metastases: A Systematic Review

**DOI:** 10.3390/diagnostics12030715

**Published:** 2022-03-15

**Authors:** Okker D. Bijlstra, Maud M. E. Boreel, Sietse van Mossel, Mark C. Burgmans, Ellen H. W. Kapiteijn, Daniela E. Oprea-Lager, Daphne D. D. Rietbergen, Floris H. P. van Velden, Alexander L. Vahrmeijer, Rutger-Jan Swijnenburg, J. Sven D. Mieog, Lioe-Fee de Geus-Oei

**Affiliations:** 1Leiden University Medical Center, Department of Surgery, Albinusdreef 2, 2333 ZA Leiden, The Netherlands; m.m.e.boreel@lumc.nl (M.M.E.B.); a.l.vahrmeijer@lumc.nl (A.L.V.); j.s.d.mieog@lumc.nl (J.S.D.M.); 2Department of Surgery, Amsterdam University Medical Centers, Cancer Center Amsterdam, University of Amsterdam, 1081 HV Amsterdam, The Netherlands; r.j.swijnenburg@amsterdamumc.nl; 3Section of Nuclear Medicine, Department of Radiology, Leiden University Medical Centre, 2333 ZA Leiden, The Netherlands; s.van_mossel@lumc.nl (S.v.M.); d.d.d.rietbergen@lumc.nl (D.D.D.R.); f.h.p.van_velden@lumc.nl (F.H.P.v.V.); l.f.de_geus-oei@lumc.nl (L.-F.d.G.-O.); 4Section of Interventional Radiology, Department of Radiology, Leiden University Medical Center, Albinusdreef 2, 2333 ZA Leiden, The Netherlands; m.c.burgmans@lumc.nl; 5Department of Medical Oncology, Leiden University Medical Center, Albinusdreef 2, 2333 ZA Leiden, The Netherlands; h.w.kapiteijn@lumc.nl; 6Department of Radiology and Nuclear Medicine, Amsterdam University Medical Center, 1081 HV Amsterdam, The Netherlands; d.oprea-lager@amsterdamumc.nl; 7Biomedical Photonic Imaging Group, University of Twente, 7522 NB Enschede, The Netherlands

**Keywords:** positron emission tomography, colorectal cancer, colorectal liver metastases, follow-up

## Abstract

(1) Background: Up to 50% of patients with colorectal cancer either have synchronous colorectal liver metastases (CRLM) or develop CRLM over the course of their disease. Surgery and thermal ablation are the most common local treatment options of choice. Despite development and improvement in local treatment options, (local) recurrence remains a significant clinical problem. Many different imaging modalities can be used in the follow-up after treatment of CRLM, lacking evidence-based international consensus on the modality of choice. In this systematic review, we evaluated ^18^F-FDG-PET-CT performance after surgical resection, thermal ablation, radioembolization, and neoadjuvant and palliative chemotherapy based on current published literature. (2) Methods: A systematic literature search was performed on the PubMed database. (3) Results: A total of 31 original articles were included in the analysis. Only one suitable study was found describing the role of ^18^F-FDG-PET-CT after surgery, which makes it hard to draw a firm conclusion. ^18^F-FDG-PET-CT showed to be of additional value in the follow-up after thermal ablation, palliative chemotherapy, and radioembolization. ^18^F-FDG-PET-CT was found to be a poor to moderate predictor of pathologic response after neoadjuvant chemotherapy. (4) Conclusions: ^18^F-FDG-PET-CT is superior to conventional morphological imaging modalities in the early detection of residual disease after thermal ablation and in the treatment evaluation and prediction of prognosis during palliative chemotherapy and after radioembolization, and ^18^F-FDG-PET-CT could be considered in selected cases after neoadjuvant chemotherapy and surgical resection.

## 1. Introduction

Colorectal cancer (CRC) is the third most common and the second most lethal cancer worldwide. In 2020, the estimated cases of CRC were 1.9 million, with 0.9 million deaths. Incidence of CRC in 2040 is predicted to increase to 3.4 million cases due to population aging [1]. Metastases occur most commonly in the liver. Up to 50% of patients with colorectal cancer either have synchronous colorectal liver metastases (CRLM) or develop CRLM over the course of their disease [2]. For curative treatment, surgical resection is the treatment of choice. However, a large number of patients are deemed irresectable at diagnosis. Anatomical variants, size and location of the lesions, the number of lesions, the number of segments involved, and the presence of extrahepatic disease affect the possibility of (curative) resection. A second treatment option with curative intent is thermal ablation (i.e., radiofrequency ablation (RFA) and microwave ablation (MWA)), which uses hyperthermia to induce tumor cell death. These local treatment methods are less invasive than surgery and serve as an alternative for patients whose condition does not permit resection or when the location of the metastasis is less suitable for surgical resection [3]. Neoadjuvant chemotherapy is mostly used as preoperative treatment in potentially locally treatable disease, although it is not routinely used as a neoadjuvant treatment option prior to resection. If surgical treatment and thermal ablation are not directly feasible, chemotherapy could be used as a neoadjuvant treatment option to provide a bridge to surgery. Frequently prescribed chemotherapy agents include fluorouracil (5-FU), leucovorin, capecitabine, oxaliplatin, bevacizumab, and irinotecan [2]. Often, a combination of chemotherapeutics is used [4]. When local treatment options are not possible and/or neoadjuvant chemotherapy has failed, radioembolization (RE) with yttrium-90 (^90^Y)- or holmium-166 (^166^Ho)-loaded microspheres has been used increasingly for palliative treatment purposes [5,6]. For more advanced (extrahepatic) disease/inoperable CRLM, chemotherapy may be prescribed as palliative treatment.

Despite development and improvement in local treatment options, unsuccessful treatment remains a significant clinical problem, leading to local and/or distant tumor recurrence. Many different imaging modalities can be used in the follow-up after treatment of CRLM. Currently, contrast-enhanced computed tomography (ceCT) is the recommended imaging modality based on the European Society of Medical Oncology (ESMO) guidelines [6,7]. Angiogenic alterations can be visualized after administration of intravenous contrast ceCT and can be indicative for various malignant processes. Changes in tumor perfusion may appear before pathologic morphological alterations in response to therapy. However, lesions <10 mm can be missed on CT and may be difficult to distinguish from benign lesions [8]. Magnetic resonance (MR) imaging is also widely used in the evaluation of CRLM because of its superiority in soft tissue resolution, multiparametric tissue characterization, and detection of subcentimetric lesions [9]. The use of diffusion-weighted (DWI) MR imaging and liver-specific contrast agents have further improved the sensitivity and specificity for intrahepatic lesion detection [10]. DWI is a functional MR technique that measures the Brownian motion of water molecules in biological tissues, which is restricted by an increase in cellularity and architectural tissue changes. Consequently, water diffusion properties are altered in tumors because of the coexistence of dense cellularity, fibrosis, necrosis, neovascularization, and hemorrhage. This results in a higher apparent diffusion coefficient (ADC). Positron-emission tomography (PET) using the radiotracer ^18^F-fluorodeoxyglucose (^18^F-FDG) can be valuable in the differentiation of treatment-induced morphological changes and residual viable tumor tissue after local curative-intent treatment (i.e., surgery and or thermal ablation) and to assess pathological response during or after neoadjuvant chemotherapy.

In a palliative setting, (early) evaluation of response to treatment of CRLM is crucial as it allows for early switching of therapy and predicts survival. Response to treatment as assessed on ceCT and MR imaging is determined by the Response Evaluation Criteria in Solid Tumors (RECIST) and Choi criteria (Table 1). However, morphological changes in the liver after radioembolization (RE) and palliative chemotherapy may hamper accurate treatment evaluation and restaging of CRLM based on RECIST and/or Choi criteria. ^18^F-FDG-PET-CT can also be used in the treatment evaluation after RE and response to palliative chemotherapy [8]. The glucose analogue ^18^F-FDG is taken up by malignant cells in which short-term tumor response can be evaluated. ^18^F-FDG activity is commonly semi-quantitively assessed by the maximum standardized uptake value (SUV_max_). The SUV value is a semiquantitative measurement of tracer uptake in a manually or semiautomatically defined volume of interest (VOI) normalized to the administered dose of the radiotracer and the total body weight or lean body mass, and is positively correlated with tumor metabolic activity [11]. Other semiquantitative clinical parameters used in PET are peak SUV (SUV_peak_), mean SUV within the tumor (SUV_mean_), metabolic (active) tumor volume (MTV), and total lesion glycolysis (TLG, a product of SUV_mean_ and MTV) [12,13,14,15]. The low spatial resolution of PET and the physiological uptake of liver parenchyma make it difficult to identify small lesions. The initially, relatively high costs compared with ceCT alone and additional exposure to radiation were factors delaying the clinical implementation of ^18^F-FDG-PET-CT for CRLM. The current new-generation PET/CT scanners, however, have increased sensitivity, allowing administration of a lower dose of the radiotracer and/or higher patient throughput, which reduces radiation exposure and costs. Moreover, standardized criteria for image interpretation and documentation in the treatment evaluation of systemic therapy and RE were lacking until recently. Similar to RECIST and Choi criteria, the PERCIST and EORTC PET criteria were developed for an objective evaluation after systemic treatment of CRLM [16] (Table 1).

There is a lack of evidence-based international consensus and there are inconsistent results on the use of the imaging modality of choice in the posttreatment evaluation after both curative-intent and palliative treatment of CRLM. Therefore, the ability of ^18^F-FDG-PET-CT in the detection of residual CRLM after surgery, thermal ablation, and neoadjuvant chemotherapy was evaluated in this systematic review. Second, the accuracy of ^18^F-FDG-PET-CT in the evaluation of response to treatment and prediction of survival after palliative chemotherapy and RE was studied. In the first part of this systematic review, the role of ^18^F-FDG-PET-CT after curative-intent treatment is summarized; secondly, ^18^F-FDG-PET-CT performance during and after palliative treatment options is evaluated. Finally, a number of highlights regarding future perspectives of PET imaging are addressed in the third section.

## 2. Method

The PubMed online database was searched on 8 July 2021. Keywords of the search included “colorectal neoplasms”, “neoplasm metastases”, “positron emission tomography computed tomography”, and “follow-up”. Articles were included when: (1) patients were treated for colorectal liver metastases; (2) the value of ^18^F-FDG-PET(-CT) was evaluated in the follow-up after local or systematic treatment; (3) manuscripts were available in English; systematic reviews, reviews, conference abstracts, and meta-analyses were excluded from analysis in this manuscript. Studies that included less than 10 patients and studies not discriminating CRLM from primary liver tumors and other liver metastases were also excluded.

Reviewing was performed independently by two authors (O.D.B. and M.M.E.B.) during the entire process to decrease the risk of selection bias, following PRISMA guidelines [20]. In case of discrepancies between the two readers, the manuscripts were evaluated again by both readers to achieve consensus.

## 3. Results

A total of 367 studies, published December 2004 and before July 2021 were included for initial identification. After filtering for “in-human studies” only, 361 manuscripts were selected for screening purposes. Both O.D.B. and M.M.E.B. screened all manuscripts and based on title and abstract agreed on excluding 304 articles for various reasons (e.g., studies focused on: imaging prior to treatment, imaging of primary colorectal tumors, no PET-CT in title, reviews, conference abstracts, meta-analyses and case reports). It was unanimously decided to read 57 full-text articles. Both reviewers selected finally the same 24 articles, and consensus was reached for 6 other articles, resulting in 30 manuscripts for final inclusion in this systematic review, which were subdivided per treatment strategy: 1 article focused solely on ^18^F-FDG-PET-CT performance after surgery, 7 articles focused on RFA and/or MWA, 9 articles studied the value of ^18^F-FDG-PET-CT after neoadjuvant chemotherapy, 6 articles analyzed ^18^F-FDG-PET-CT during and after palliative chemotherapy, and 7 articles were included on ^18^F-FDG-PET-CT in the follow-up after radioembolization. Reasons for exclusion and the review process are summarized in the PRISMA flow chart, displayed in Figure 1.

### 3.1. ^18^F-FDG-PET-CT Performance after Surgical Resection

The main aims of diagnostic follow-up after local treatment of CRLM are early non-invasive detection of residual tumors and local tumor progression and the detection of new intrahepatic metastases and extrahepatic disease. Anatomical imaging by ceCT is the recommended imaging modality based on the ESMO guidelines and is therefore traditionally the most frequently performed diagnostic imaging modality for patients suffering from resectable CRLM. MR imaging may be superior to ceCT for the early detection of local tumor progression, but it is less suitable for detecting extrahepatic disease. The caveat of PET/MR imaging is that it yields decreased sensitivity for detecting lung metastases, for which CT is the modality of choice. The adoption of integrated ^18^F-FDG-PET-CT combines anatomical and metabolic imaging. The addition of ^18^F-FDG-PET provides complementary metabolic information that enables the detection of malignant disease at unexpected sites or in morphologically normal structures that may be easily overlooked on morphological imaging [8,21,22].

Only one suitable study was found to have studied the role of ^18^F-FDG-PET imaging after surgical resection of CRLM. Vigano et al. [23] compared ^18^F-FDG-PET-CT with CT or MR imaging in 107 patients with recurrence of CRLM after liver resection. These patients were rediscussed in a multidisciplinary team meeting after an additional PET-CT. Sensitivity of local liver recurrences for CT and MR imaging and ^18^F-FDG-PET-CT were 100% and 96.7%, respectively. In comparison with CT or MR imaging, 24 additional, extrahepatic, malignant sites were discovered by ^18^F-FDG-PET-CT. ^18^F-FDG-PET-CT, therefore, altered treatment strategy in 16 patients. Fifteen patients did not undergo surgery due to extrahepatic disease, which was only detected on ^18^F-FDG-PET-CT.

#### Summary

Although only one study was published specifically on the ^18^F-FDG-PET-CT performance after surgical resection of CRLM, ^18^F-FDG-PET-CT seems to have an additional value for the detection of extra hepatic recurrences (mainly lymph node metastases, bone metastases, and peritoneal carcinomatosis). ^18^F-FDG-PET-CT has low false positive and false negative rates (3.1% and 1.3%, respectively) in detecting extrahepatic disease. In the detection of local recurrences in the liver, additional ^18^F-FDG-PET-CT, however, does not seem to contribute to follow-up after resection compared with ceCT alone [24,25].

### 3.2. ^18^F-FDG-PET-CT after Thermal Ablation

The additional metabolic information provided by ^18^F-FDG-PET may be especially useful after local treatment with thermal ablation since treatment-induced changes are difficult to differentiate from residual viable tumor tissue on morphological imaging. Here the role of ^18^F-FDG-PET-CT after thermal ablation is summarized. ^18^F-FDG-PET-CT timing can be divided in three categories: (1) immediate ^18^F-FDG-PET-CT (i.e., within minutes after completion of ablation); (2) early ^18^F-FDG-PET-CT (i.e., within 24–48 h after ablation); and (3) follow-up ^18^F-FDG-PET-CT (i.e., weeks–months post-treatment).

In a small retrospective cohort study in 11 patients with 16 CRLM, pre- and post-ablation PET-CTs were made for the detection of residual tumor after RFA. Post-ablation scans were made within 48 h after treatment [26]. ^18^F-FDG-PET and ^18^F-FDG-PET-CT accuracy was 68% for detecting residual disease within 48 h, and ^18^F-FDG-PET-CT identified five local recurrences in four patients during later follow-up that were not found on ceCT, leading to earlier reintervention. Local recurrence in the ablation zone occurred in six patients during later follow-up (mean follow-up 393 days) despite negative early ^18^F-FDG-PET-CT.

Kuehl et al. [27] compared ^18^F-FDG-PET, ^18^F-FDG-PET-CT and MR imaging performance for the detection of local recurrence after RFA in 16 patients. All patients received a baseline ^18^F-FDG-PET-CT 48 h prior to ablation, immediately after RFA, and at follow-up (1, 3, and every 6 months post-RFA). Confirmation by histology or ceCT combined with clinical parameters served as reference standard. Focal uptake on ^18^F-FDG-PET in the ablation zone was seen as local recurrence. Local recurrence on contrast-enhanced (gadolinium) MR imaging was concluded when morphologically detectable tumor (hypointense lesion on T1 sequence, hyperintense on T2) was found in or around (1 cm) the ablation zone. In total, ^18^F-FDG-PET-CT missed 13 lesions, of which 4 were missed at the 24 h post-RFA scan, and 5 were missed at 1 month post-RFA. MR imaging missed eight lesions of which three were missed 24 h post-RFA and at 3 months, and two were missed at 1 month after RFA. No significant difference in diagnostic value was found between ^18^F-FDG-PET-CT and MR imaging over the course of the follow-up.

In a large retrospective cohort study by Sahin et al. [28] ^18^F-FDG-PET performance was analyzed in 134 patients with ^18^F-FDG-PET scans available before laparoscopic thermal ablation. Post-ablation follow-up ^18^F-FDG-PET-CT scans (the timing of ^18^F-FDG-PET-CT varied) were requested and made at the discretion of the surgeon or the oncologist in 82 patients with 180 lesions; 72% of these patients showed rising serum CEA levels. Follow-up ^18^F-FDG-PET-CT performance was superior to ceCT in 11 of 51 patients (22%) and inferior to ceCT in 2 of 51 patients (4%) diagnosed with local recurrence.

Liu et al. [29] studied the efficacy of early ^18^F-FDG-PET-CT scanning after percutaneous RFA in 12 patients with 20 suspected lesions. All patients received an ^18^F-FDG-PET-CT within two weeks prior to RFA, within 24 h post-ablation, and at 1-, 3-, and 6-month follow-up. Clinical and radiological follow-up was considered as the reference standard. A lesion was considered local recurrence when showing corresponding morphology on CT and/or metabolic activity on ^18^F-FDG-PET imaging. Three out of twenty lesions showed local recurrence on early ^18^F-FDG-PET-CT. All early ^18^F-FDG-PET-CT findings corresponded to results during 1-, 3-, and 6-month follow-up. No comparison was made with other imaging modalities.

Nielsen et al. [30] included 79 patients with a total of 179 RFA-treated lesions and studied ^18^F-FDG-PET-CT performance for detecting local recurrence. Local recurrence was classified when ^18^F-FDG-PET showed focally increased FDG uptake in the original tumor periphery that could not be correlated with inflammation (i.e., a rim-shaped enhancement pattern). Follow-up (imaging) data or histopathology was chosen as reference standard. They found local recurrence in 30 of 79 (38%) patients, which were all detectable by ^18^F-FDG-PET imaging; local recurrence was missed by ceCT in three patients (10%).

Cornelis and colleagues [31] examined the ability of immediate ^18^F-FDG-PET-CT and ceCT to predict local treatment failure 1 year after thermal ablation in 21 patients. Patients received a split-dose of ^18^F-FDG (i.e., one-third of the standard dose prior to ablation and a second dose, equivalent to two-thirds of the average dose, upon completion of the ablation procedure) for intraprocedural guidance and direct postprocedural assessment of treatment success. ^18^F-FDG-PET-CT scans were acquired just before and directly after the ablation with the patient positioned in the same bed position. Recurrence was predicted accurately on ^18^F-FDG-PET-CT with sensitivity and specificity of 100% and 85.7%, respectively.

In another study performed by Cornelis et al. [32], the value of an immediate ^18^F-FDG-PET-CT after thermal ablation of 62 ablation zones in 39 patients using the same split-dose protocol was evaluated. SUV ratios were calculated in post-ablation ^18^F-FDG-PET-CTs and were correlated with histopathological analysis of biopsies. The SUV ratio was calculated as follows: (VOI ablation zone SUV_max_—VOI healthy liver SUV_mean_)/VOI healthy liver SUV_mean_. Most patients (74%) received MWA, and median follow-up time was 22.5 months. Local tumor progression was found in 37% of treated tumors. Significantly higher SUV ratios were found in tumors developing local recurrence compared with adequately treated CRLM.

#### Summary

^18^F-FDG-PET-CT seems to play an important role as a diagnostic imaging modality in the follow-up after thermal ablation of CRLM. In this regard, two strategies can be distinguished: early post-ablation ^18^F-FDG-PET-CT (i.e., within 24–48 h) and follow-up ^18^F-FDG-PET-CT 1–3 months after treatment. In general, independent of early (within 48 h) imaging, multiple studies show a clinical superiority of ^18^F-FDG-PET-CT over ceCT in detecting local tumor progression after thermal ablation. The improved accuracy of follow-up imaging underlines the incremental interest for minimally invasive local ablation therapy. The increased sensitivity resulted in the detection of smaller tumors, which are more amenable to local treatment and which can postpone systemic therapy. An example of the value for detecting residual disease 3 months after ablations is provided in Figure 2, and an overview of included studies and results is summarized in Table 2.

### 3.3. ^18^F-FDG-PET-CT as Response-Monitoring Modality after Neoadjuvant Chemotherapy

Multiple studies have been conducted to evaluate the performance of ^18^F-FDG-PET-CT after chemotherapy, both in neoadjuvant and palliative settings. Predicting pathological response to treatment is important during and after neoadjuvant chemotherapy to determine (early) treatment response and for the optimal timing of local curative treatment. Different outcome measures of PET-CT were compared with various diagnostic methods in the treatment response evaluation of chemotherapy in CRLM, and many different chemotherapeutics were used. In this chapter, ^18^F-FDG-PET-CT performance for response evaluation after neoadjuvant chemotherapy is described. Additionally, studies may be divided into three categories by timing of ^18^F-FDG-PET-CT imaging: (1) early treatment response monitoring; (2) mid-treatment response monitoring; and (3) end-of-treatment response evaluation prior to local treatment.

Sensitivity and specificity of ^18^F-FDG-PET-CT were compared with ceCT imaging in 48 patients after receiving neoadjuvant chemotherapy and in 27 patients without neoadjuvant chemotherapy, prior to surgery [33]. Sensitivity and specificity of ^18^F-FDG-PET-CT for the detection of CRLM were significantly lower in the group receiving neoadjuvant treatment, and ceCT showed higher sensitivity and specificity in both groups compared with ^18^F-FDG-PET-CT.

The role of standardized added metabolic activity (SAM) measurement in ^18^F-FDG-PET-CT was studied prospectively in 18 patients receiving neoadjuvant chemotherapy before surgery [34]. Patients were classified into one of four response categories, according to RECIST criteria. The SAM was calculated by drawing a VOI (VOI_1_) and a larger VOI (VOI_2_) around VOI_1_ and subsequently subtracting background signal in (VOI_2_-VOI_1_) from the background in VOI_1_. In contrast with assessment according to RECIST, SUV_max_ and SAM differed significantly between responders and non-responders. High follow-up SUV_max_ and low ΔSAM were significantly correlated with worse PFS and OS; however, ΔSUV_max_ was no predictor of PFS and OS. No correlation of metabolic and pathologic response was described in this study.

^18^F-FDG-PET-CT was compared with superparamagnetic iron oxide-enhanced (SPIO) MR imaging in 19 patients after completion of neoadjuvant chemotherapy by Bacigalupo et al. [35] SPIO-MR imaging detected 125 out of the 136 metastases, where ^18^F-FDG-PET-CT detected only 71 lesions. Metastases were confirmed with intraoperative ultrasound and/or pathology results as reference standard. Differences in sensitivity were detected in lesions <15 mm and lesions between 15 and 30 mm. In lesions >30 mm, no difference in sensitivity was observed between either imaging strategy. No data on the exact timing of imaging after chemotherapy and the time interval between imaging and surgery were reported.

Garcia Vincente et al. [36] specifically described the role of ^18^F-FDG-PET-ceCT. Nineteen patients with CRLM were evaluated with ^18^F-FDG-PET-CT, ^18^F-FDG-PET, and ceCT after completion (i.e., after four cycles) of neoadjuvant chemotherapy. A total of 105 CRLM were detected after neoadjuvant chemotherapy. Histology was chosen as reference standard when patients underwent resection after neoadjuvant chemotherapy, and evaluation by a multidisciplinary team meeting was considered as the reference standard when a patient did not undergo surgery. ROC analysis showed values of 0.691 (*p* = 0.149), 0.957 (*p* = 0.001), and 0.974 (*p* < 0.005) for ^18^F-FDG-PET, ceCT, and ^18^F-FDG-PET-CT, respectively. Additionally, a significant correlation was found between lesion size and ceCT and ^18^F-FDG-PET performance. A stratified analysis was performed for the lesions on ^18^F-FDG-PET greater than 10 mm and lesions smaller than 10 mm. A non-significant higher sensitivity was found for lesions >10 mm.

Burger et al. [37] studied 69 patients who received an ^18^F-FDG-PET-CT between 2 and 7 weeks after neoadjuvant chemotherapy (within 8 weeks prior to surgery). Change in SUV (ΔSUV) before and after chemotherapy was compared with a histopathological tumor regression grade (TRG). In TRG 1–3, no viable tumor cells to maximum 50% tumor cells were present; in TRG 4–5, tumor cells were found in 50–100% of the histological specimen. A significant correlation between ΔSUV and TRG with an area under the curve (AUC) of 0.773 was found. An optimal cut-off point of 41% ΔSUV was measured to distinguish responders (TRG 1–3) from non-responders (TRG 4–5).

The predictive value of ^18^F-FDG-PET-CT for pathologic response to neoadjuvant chemotherapy was assessed in 34 patients by Nishioka et al. [38]. ^18^F-FDG-PET-CT scans were made within 1 month prior to surgery, and only 10 patients also received an ^18^F-FDG-PET-CT scan before initiation of chemotherapy. A moderate correlation (r = 0.660, *p* < 0.001) was found between the SUV_mean_ and tumor viability in general. However, when low SUV_mean_ and low SUV_max_ were correlated with a tumor viability lower than 10%, both parameters showed to be strong predictors of tumor viability ≤10% (AUC 0.916 and AUC 0.887, respectively).

A direct comparison between metabolic response measured on ^18^F-FDG-PET-CT and pathologic response as assessed after resection in 14 patients receiving neoadjuvant chemotherapy was made by Tan et al. [39]. Of the 34 lesions that showed complete metabolic response on ^18^F-FDG-PET-CT, 29 (85%) lesions still had viable tumor cells on pathology assessment.

In a small prospective analysis by De Bruyne et al. [40], the predictive value of dynamic contrast-enhanced MR imaging (DCE-MR imaging) and ^18^F-FDG-PET-CT was analyzed before and after completion (i.e., after five cycles) of neoadjuvant treatment. MR imaging was evaluated by RECIST, and ^18^F-FDG-PET-CT imaging was evaluated following EORTC criteria. Baseline parameters on DCE-MR imaging were not predictive for overall survival (OS) and progression-free survival (PFS). Baseline SUV_max_ did not differ between the group of responders and the group of non-responders. Decrease in SUV_max_ at follow-up, however, was correlated with increased PFS. No correlation of the change in SUV_max_ (ΔSUV_max_) and pathologic response was performed in this analysis.

Lastoria et al. [41] imaged 33 patients prior to chemotherapy and after one cycle of neoadjuvant chemotherapy. Metabolic response was evaluated with SUV_max_ and TLG. Compared with RESIST criteria, measured with CT imaging, SUV_max_ and TLG had superior predictive values for PFS and OS.

#### Summary

The analyzed studies used different treatment regimes, imaging timepoints, and assessment criteria. Three studies assessed metabolic response on ^18^F-FDG-PET-CT using SUV_mean_, SUV_max_, and ΔSUV as predictors of pathologic response and concluded poor-to-moderate predictive performance. Multiple parameters such as SUV_max,_ MTV, and TLG seem to have a predictive value on patient outcomes and could be used as a prognostic marker for the prediction of long-term outcomes in CRLM treated by neoadjuvant chemotherapy and surgery. Accordingly, the vast majority of the evaluated articles have positive outcomes on the performance of ^18^F-FDG-PET-CT after neoadjuvant chemotherapy for CRLM. Typical findings on ^18^F-FDG-PET-CT before and after neoadjuvant therapy is displayed in Figure 3, and an overview of included studies and results is summarized in Table 3.

### 3.4. ^18^F-FDG-PET-CT as Response Monitoring during and after Palliative Chemotherapy

Response monitoring during palliative chemotherapy is also pivotal because some patients initially deemed irresectable may show a good response to chemotherapy, making them suitable for curative-intent local therapy. Moreover, response monitoring of palliative chemotherapy is useful when deciding which chemotherapeutic agent is best suitable for an individual patient. Early evaluation of response to treatment enables the physician to switch to another chemotherapeutic regimen early during treatment or, if possible, to temporarily halt chemotherapy.

Heijmen et al. [42] assessed the value of ^18^F-FDG-PET-CT after three cycles of chemotherapy in 39 patients. A total of 5 patients received neoadjuvant chemotherapy, and 35 patients received palliative chemotherapy. Prechemotherapy and postchemotherapy SUV_max_ and TLG on ^18^F-FDG-PET-CT and ADC and T2*MR on T1.5 MR images were measured to predict the response to systemic treatment. A high SUV_max_, TLG, low ADC, and high T2*MR prior to treatment were correlated with a shorter OS. Low ADC before treatment was associated with shorter PFS. A decrease in SUV_max_ and increase in ADC was seen after one week of chemotherapy. These parameters were significantly correlated with each other, but were not predictive of OS or PFS.

In a prospective phase II trial, response evaluation based on ^18^F-FDG-PET-CT PERCIST criteria was compared with RECIST criteria on ceCT in 61 patients receiving palliative treatment (cetuximab –irinotecan) [43]. Imaging was performed within 2 weeks prior to the initiation of treatment and after every four cycles of chemotherapy. None of the patients reached complete response based on ceCT and ^18^F-FDG-PET-CT, 11 patients (18%) had partial response (RECIST) compared with partial metabolic response in 34 patients (56%) according to PERCIST criteria. OS was significantly longer for patients considered as partial metabolic response compared with patients with stable metabolic disease. No significant difference in OS was observed between patients in the partial response and stable disease groups, based on RECIST.

Nemeth et al. [44] prospectively studied the relation of metabolic changes on ^18^F-FDG-PET-CT to PFS in 53 patients after two cycles (8 days after the second cycle) of combined chemotherapy. A total of 10 out of 53 patients received neoadjuvant chemotherapy prior to liver resection during the study period, and 43 patients were treated with palliative chemotherapy. Metabolic response was assessed according to adapted EORTC criteria. Baseline and percentage change (Δ) in SUV_max_, TLG, SAM, and normalized SAM (NSAM) were calculated. SAM and NSAM were both correlated with PFS and OS, whereas neither SUV_max_ and TLG nor ΔSUV_max_ and ΔTLG were predictors of PFS and OS.

In a retrospective analysis in 40 patients receiving palliative chemotherapy, the correlation of a complete metabolic response on ^18^F-FDG-PET-CT and PFS and OS was studied [45]. The authors found that patients achieving complete metabolic response had improved PFS and OS. Moreover, patients with low baseline SUV_max_ were more likely to maintain complete metabolic response.

To predict early tumor response, Hyun Kim et al. [46] evaluated 17 patients who underwent ^18^F-FDG-PET-CT before and after the first cycle of chemotherapy. Non-responders after 1 cycle of chemotherapy were assigned to second-line or third-line chemotherapy. Different baseline values and reduction rates of the parameters of ^18^F-FDG-PET-CT and 3D perfusion CT were compared between responders and non-responders. Significant differences between responders and non-responders were found in reduction rates of 30% of MTV and TLG on ^18^F-FDG-PET-CT. On perfusion, CT blood flow and the flow extraction product showcased a higher decrease in responders compared with non-responders.

Correa-Gallego et al. [47] evaluated the use of ^18^F-FDG-PET-CT after hepatic arterial infusion pump (HAIP) with chemotherapy. A baseline ^18^F-FDG-PET-CT and a ^18^F-FDG-PET-CT after three and six cycles of induction chemotherapy was made in 49 patients with unresectable CRLM. Metabolic response was measured with ΔSUV_max_, ΔSUV_mean_, and ΔTLG. Outcome measures were conversion to resection, OS, PFS, and disease-free survival (DFS). ΔSUV_max_ and ΔTLG were not higher in patients who were deemed resectable after chemotherapy compared with patients who could not undergo resection. Moreover, metabolic parameters did not correlate with OS, PFS, and DFS and could not be used as prognostic parameters in the response to HAIP-chemotherapy. The authors suggest the poor performance of ^18^F-FDG-PET-CT in HAIP-chemotherapy may be a result of decreased hexokinase activity as a result of the chemotherapy’s hepatotoxicity.

#### Summary

^18^F-FDG-PET-CT seems to be a suitable imaging technique for assessing response to treatment during and after palliative chemotherapy. Multiple studies have shown the predictive value of metabolic response evaluation on ^18^F-FDG-PET-CT for PFS and OS, whereas only one study evaluating chemotherapy via HAIP concluded that ^18^F-FDG-PET-CT was not predictive of PFS and OS. Both traditional metabolic parameters (SUV_max_ and TLG) and more complex parameters (SAM and NSAM) are suitable for response evaluation and can be useful for prediction of PFS and OS. An overview of included studies and results is summarized in Table 4.

### 3.5. ^18^F-FDG-PET-CT after Radioembolization

Zerizer et al. [17] compared EORTC PET criteria with RECIST and Choi criteria in 25 patients with 121 CRLM 6–8 weeks after treatment with ^90^Y-RE. Imaging parameters were correlated with changes in tumor markers and 2-year PFS rates. Significantly more patients showed partial response to treatment according to PET criteria compared with both RECIST and Choi criteria. Moreover, metabolic response on ^18^F-FDG-PET-CT imaging was strongly correlated with normalization of tumor markers and was a better predictor of PFS.

The efficacy of RE evaluated by ^18^F-FDG-PET-CT parameters was studied by Soydal et al. [48] in 2013 in 35 patients. They calculated MTV pre- and posttreatment and the change in TLG six weeks after treatment. A ΔTLG of 26.5 was calculated as the cut-off for responders and non-responders. Mean survival for responders was 20.76 (±2.71) months and 11.32 (±1.18) months for non-responders, respectively.

In a German study performed by Sabet et al. [49], the predictive value of metabolic response 4 weeks after RE on ^18^F-FDG-PET-CT was evaluated in 51 patients. Three CRLMs with the highest SUV_max_ were identified as target lesions in every patient; a 50% decrease in tumor-to-background ratio was seen as metabolic response. Early metabolic responders, i.e., 4 weeks after treatment, had significantly longer OS than non-responders (10 months vs. 4 months, respectively).

RECIST 1.1, tumor attenuation criteria, Choi criteria, and EORTC PET criteria were studied in a retrospective analysis in 25 patients (46 target lesions) for response assessment and prediction of hepatic PFS after RE by Shady et al. [50]. A statistically significant correlation between a change in tumor attenuation, measured in HU, and SUV_max_ change was seen. Moreover, assessments following Choi criteria, tumor attenuation, and EORTC PET criteria were found to be predictors of hepatic PFS.

A retrospective study including 49 patients with 119 target CRLM compared the RECIST criteria with metabolic response on ^18^F-FDG-PET-CT, based on SUV_max_, SUV_peak_, MTV, and TLG for prediction of treatment response and OS [51]. Response assessment by MTV and TLG showed a statistically significant correlation with prediction of OS. Response measured with SUV_max_ and SUV_peak_ and no progression of disease based on RECIST were not associated with prolonged OS.

In a study including 38 CRLM patients treated with RE, anatomic response assessed with MR imaging was compared with metabolic tumor response evaluated with ^18^F-FDG-PET-CT [52]. All patients received a baseline MR imaging and ^18^F-FDG-PET-CT prior to treatment and at 1 and 3 months, respectively, after treatment with ^90^Y microspheres. Anatomic response assessment on MR imaging was performed following RECIST 1.1 criteria with longest tumor diameter (LTD) as variable, and the metabolic response was quantitatively assessed based on total liver TLG (sum of all TLG values). One month after treatment, objective response rates were 11% and 39% for RECIST and total liver TLG, respectively. At three months after treatment, the objective response rates were 24% and 33% for RECIST and total liver TLG, respectively. Moreover, a decrease in total liver TLG at 1 and 3 months after treatment was associated with an increase in OS; LTD reduction was found to be significantly correlated with longer OS only 3 months after treatment. Thus ^18^F-FDG-PET-CT enabled response prediction 1 month after treatment, whereas prediction based on CT was only possible 3 months after treatment.

Sager et al. [53] compared RECIST and PERCIST criteria after ^90^Y RE in 19 patients with a total of 42 CRLM, with therapy response as the primary outcome measurement for the evaluation of treatment response. A total of 12 out of 42 lesions (29%) were categorized as partial response, 14 (33%) as progressive disease, and 16 (38%) as stable disease according to PERCIST criteria. Comparable results were reported for RECIST criteria. Although patients with progressive disease based on RECIST and PERCIST criteria seemed to have impaired survival compared with responders, this survival benefit did not reach statistical significance.

#### Summary

Multiple parameters such as SUV_max,_ MTV, and TLG seem to have a predictive value on patient outcomes and could be used as a prognostic marker for the prediction of long-term outcomes in CRLM treated by radioembolization. The additional value of ^18^F-FDG-PET-CT imaging compared with ceCT imaging is shown in Figure 4, and an overview of included studies and results is summarized in Table 5.

### 3.6. Future Perspectives

#### 3.6.1. Tumor-Targeted PET Tracers

Several novel PET tracers have been studied in CRLM patients in recent years, with varying degrees of success. A pilot study in 10 patients using a prostate-specific membrane antigen (PSMA) tracer, ^68^Ga-PSMA-11, found low PET avidity in CRLM and concluded ^68^Ga-PSMA-11 not to be a suitable tracer for the detection of CRLM [54]. The nucleoside analogue 3′-fluoro-3′deoxythymidine (FLT) tracer, ^18^F-FLT, was studied in 18 CRLM patients receiving neoadjuvant chemotherapy [55]. The authors concluded that ^18^F-FLT uptake might be a predictive imaging biomarker for early treatment response after chemotherapy. However, this study was published in 2013, and no further research supporting this hypothesis has been published since.

More recently, fibroblast-activation-protein inhibitors (FAPI) have proven to be of great potential in preclinical and clinical studies as a PET tracer for various solid tumors. Fibroblast activation protein (FAP) is overexpressed by cancer-associated fibroblasts (CAFs) in several cancer types [56]. ^68^Ga-FAPI PET-CT showed higher uptake than ^18^F-FDG PET-CT in gastric, duodenal, and colorectal adenocarcinoma liver metastases (SUV_max_ 9.7 vs. 5.2) in a study performed by Pang et al. [57]. One more advantage of ^68^Ga-FAPI PET-CT over ^18^F-FDG PET-CT is the low background SUV_max_ in the liver (1.69 vs. 2.77), improving tumor-to-background ratios even further [58]. In a retrospective cohort study of 14 patients with CRLM, an SUV_max_ value of 9.54 was measured on ^68^Ga-FAPI PET-CT. Unfortunately, no comparison was made with ^18^F-FDG PET-CT [59].

Şahin and colleagues [60] performed a one-to-one comparison between ^68^Ga-DOTA-FAPI PET-CT and ^18^F-FDG PET-CT in 15 patients with CRLM. The differences in measured median SUV_max_ were 5.5 and 5.0 (*p* = 0.25) in ^68^Ga-DOTA-FAPI and ^18^F-FDG PET-CT, respectively. A significantly different median tumor-to-background ratio of 4.5 vs. 1.3 (*p* < 0.05) was found.

One drawback of PET-CT imaging with FAPI is the relatively high uptake in fibrotic and cirrhotic livers, leading to higher background signals and thus lower—and in one case even negative—tumor-to-background ratios [61] in patients with cirrhotic liver disease [62].

#### 3.6.2. PET-MR Imaging

^18^F-FDG-PET-MR imaging has been introduced as a very promising imaging technique simultaneously using the metabolic features of ^18^F-FDG-PET and the anatomic and functional features of MR imaging that display higher soft tissue contrast resolutions than CT. Moreover, MR imaging provides more than only anatomical information when combining different sequences and by using MR contrast agents. However, little is known about the potential role of ^18^F-FDG-PET-MR in the assessment of treatment success of CRLM. ^18^F-FDG-PET-MR was shown to be comparable in diagnostic accuracy of liver lesions compared with ^18^F-FDG-PET-CT imaging in several studies [63,64,65], and some studies concluded superiority of ^18^F-FDG-PET-MR over ^18^F-FDG-PET-CT [66,67,68]. This increased accuracy is mainly explained by the superior soft tissue contrast of MR imaging compared with CT. Another additional benefit of MR imaging over CT is that it does not use ionizing radiation. More recently, two large observational studies concluded that ^18^F-FDG-PET-MR improves lesion detection in several cancer types and decreases false-negative rates, although no clear subanalysis confirmed this statement specifically for patients with CRLM [69,70].

Only two reports have studied ^18^F-FDG-PET-MR performance after neoadjuvant chemotherapy. In a short retrospective case series including 15 patients receiving chemotherapy, imaging data of ^18^F-FDG PET-CT and ^18^F-FDG-PET-MR were retrospectively correlated with histology or follow-up imaging as reference standard [65]. ^18^F-FDG-PET-DWI-MR yielded significantly higher sensitivity and specificity compared with ^18^F-FDG-PET-CT for the initial detection of CRLM. No subanalysis was performed to study ^18^F-FDG-PET-DWI-MR performance in detecting local tumor recurrence and/or in evaluating residual disease. A second retrospective study including 55 patients with CRLM concluded that ^18^F-FDG-PET-MR has significantly better sensitivity in detecting residual CRLM than multidetector CT in patients who had received chemotherapy recently (i.e., within the 3 months prior to ^18^F-FDG-PET-MR). However, ^18^F-FDG-PET-MR did not outperform gadolinium MR imaging [67].

#### 3.6.3. Radiomics

Radiomics aims to extract a large number of quantitative features from medical imaging to provide clinicians with additional information from existing imaging invisible to the human eye [71]. Combining several features in a multiparametric model is challenging for the human eye and brain and is therefore not reproducible. The machine-learning approach in radiomics incorporates all clinical and imaging features available and can subsequently aid the clinician in decision making for patient-specific treatment and prediction of patient outcomes. Typical workflow for radiomics consist of: (1) standardized image acquisition; (2) manual or automated segmentation of the tumor(s); (3) extraction of a large number of imaging features (e.g., intensity, shape, texture); (4) analysis of features’ relationship to treatment efficacy and/or patient outcome [72,73].

In recent years, several studies have been published that specifically focus on radiomics in metastatic colorectal cancer, but only two studied radiomic features in PET, although not specifically describing radiomics in treatment evaluation after local therapy [74,75]. In a retrospective study including 99 patients receiving palliative chemotherapy three (local) intensity, four morphological, two intensity histogram, and one intensity–volume histogram radiomic features in PET imaging, acquired prior to chemotherapy, were analyzed and correlated with anatomical change per lesion, treatment benefit, PFS, and OS [74]. The authors concluded that tumor volume, tumor heterogeneity, and non-sphericity are negatively correlated with the benefit of treatment and subsequent survival. A second trial studied whether using radiomic PET features, 41 in total were explored, could serve as a prognostic model in 52 patients with CRLM [75]. They demonstrated the poor prognostic value of commonly used SUV parameters (SUV_peak_ and SUV_max_) compared with number of CRLM, MTV, and TLG. However, combining one or more radiomic features in a multivariate analysis increased prognostic accuracy, with hazard ratios for PFS and OS of 4.02 (95% CI 1.67–9.7) and 4.29 (95% CI 2.15–8.57), respectively.

More extensive research has been performed in the applications of radiomics in CRLM using other imaging modalities (e.g., CT and MR imaging) [73,76,77,78,79].

## 4. Discussion

In this systematic review, we studied the role of ^18^F-FDG-PET-CT for evaluation of treatment for CRLM. We focused on five main treatment modalities—i.e., surgery, thermal ablation (RFA and MWA), neoadjuvant chemotherapy, palliative chemotherapy, and radioembolization. ^18^F-FDG-PET-CT performance has been evaluated as a diagnostic imaging modality for detecting residual disease after neoadjuvant chemotherapy and thermal ablation. Remarkably, only one study was found to describe the role of ^18^F-FDG-PET-CT imaging after surgery, preventing a solid recommendation after this treatment modality. Furthermore, ^18^F-FDG-PET-CT imaging was evaluated for the assessment of treatment response and prediction of survival.

Studies on ^18^F-FDG-PET-CT scans during follow-up after RFA and/or MWA should be analyzed with caution since data on the exact timing of the follow-up moment are sometimes not available. However, in addition, these studies conclude that the metabolic changes in and around the ablation cavity are easier to interpret than subtle morphological changes, which may be present on ceCT. In general, independent of early (within 48 h) imaging or follow-up imaging (≥3 months), multiple studies show a clinical superiority of ^18^F-FDG-PET-CT over ceCT in detecting local tumor progression after thermal ablation.

^18^F-FDG-PET-CT parameters showed poor-to-moderate correlation with histopathological findings for smaller lesions and similar correlation with histopathology compared with morphological imaging techniques. For this reason, current imaging modalities, including ^18^F-FDG-PET-CT, cannot provide a reliable indication that predicts a complete (pathologic) response and thus should only be used to determine whether a patient can be treated locally after neoadjuvant therapy.

Multiple parameters such as SUV_max,_ MTV, and TLG seem to have a predictive value on patient outcomes and could be used as prognostic markers for the prediction of long-term outcomes in CRLM treated with palliative chemotherapy.

The role of ^18^F-FDG-PET-CT imaging in treatment evaluation after RE has been studied thoroughly. Based on the studies evaluated in this systematic review, ^18^F-FDG-PET-CT seems to outperform MR imaging and CT for response assessment and prediction of overall survival.

Currently, no data on treatment evaluation of ^18^F-FDG-PET-MR after surgical resection, thermal ablation, and radioembolization are available in the literature. Therefore, the exact role of ^18^F-FDG-PET-MR after local treatment of CRLM remains uncertain. A novel PET tracer that has recently received a lot of attention and is being researched intensively is FAPI. Although initial results seem promising, additional research is required before any conclusive statements can be made on the exact role FAPI-PET-CT could play in treatment evaluation of CRLM.

Some limitations of this review should be addressed. First, in all treatment categories heterogeneity in patients and patient characteristics between studies was found. Second, studies used a variety of methods and outcome measures, complicating the interpretation and comparability of the results summarized in this report. Nevertheless, to the best of our knowledge, we included the best currently available evidence to provide recommendations for the role of PET-CT imaging during follow-up after treatment of colorectal liver metastases.

## 5. Conclusions

In conclusion, the role of ^18^F-FDG-PET-CT for treatment evaluation of CRLM is strongly dependent on the treatment option chosen and the setting in which it is used. The use of ^18^F-FDG-PET-CT is not justified regularly after neoadjuvant chemotherapy, and the exact role of ^18^F-FDG-PET-CT after surgery remains unclear and needs to be further evaluated. In contrast, ^18^F-FDG-PET-CT seems superior to ceCT and MR imaging for the early detection of residual vital tumor tissue (<48 h after ablation) and local tumor progression up to 1 year after thermal ablation and for treatment response evaluation after radioembolization and palliative chemotherapy.

In addition, novel PET tracers and ^18^F-FDG-PET-MR imaging may be of significant added value in the near future. The relatively unexplored field of radiomics, although its first results are promising, is still in its infancy and may provide novel insights on the abovementioned conclusions over the next few years.

## Figures and Tables

**Figure 1 diagnostics-12-00715-f001:**
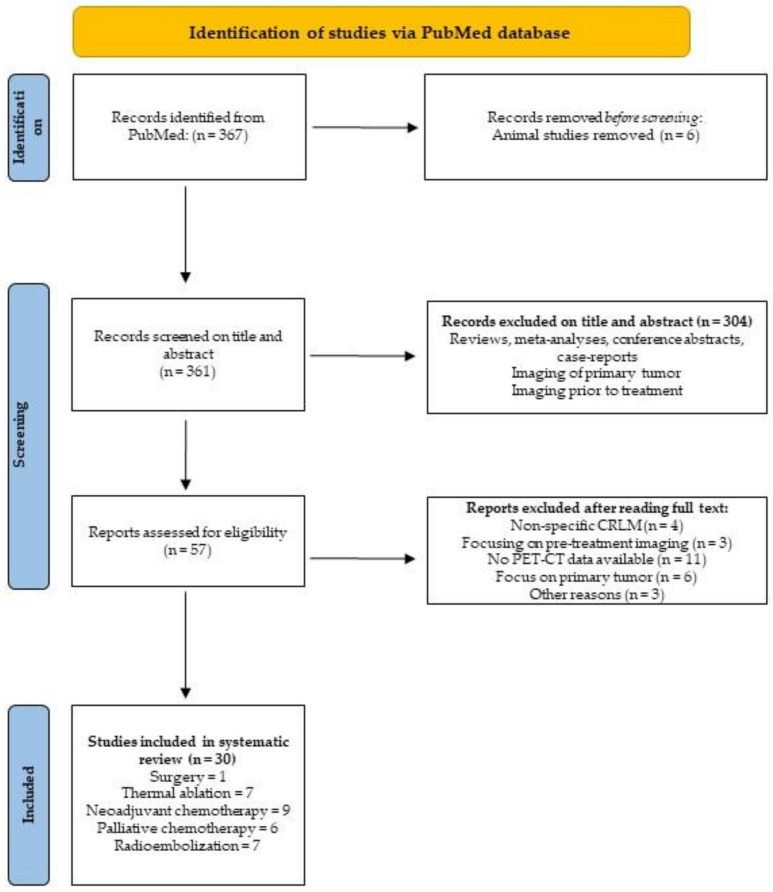
PRISMA flow chart.

**Figure 2 diagnostics-12-00715-f002:**
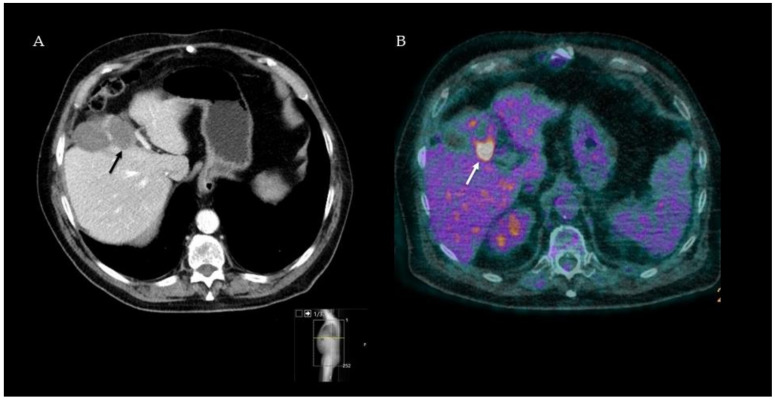
Follow-up ceCT image three months after RFA, suggesting clear ablation margins (black arrow) and no residual tumor (**A**). Simultaneous ^18^F-FDG-PET-CT image of the same patient showing high focal FDG uptake in the tumor periphery (white arrow) strongly suspected of residual disease (**B**). Three months later, CEA levels had risen, and the focal FDG uptake had spread, confirming tumor residue at the ablation site.

**Figure 3 diagnostics-12-00715-f003:**
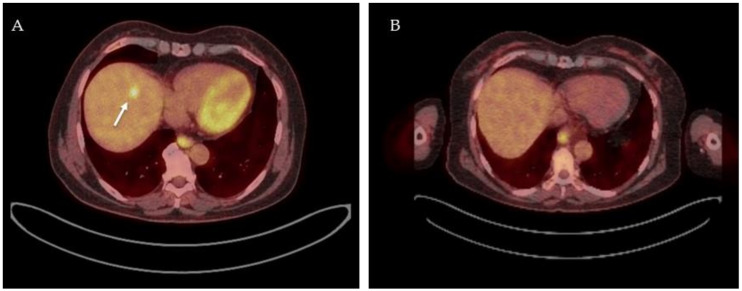
^18^F-FDG PET-CT images of a patient before and after receiving neoadjuvant chemotherapy. ^18^F-FDG PET-CT fusion image with high focal FDG uptake in segment 4A indicative of a colorectal liver metastasis, white arrow (**A**); ^18^F-FDG PET-CT fusion image after 3 cycles of neoadjuvant chemotherapy (FOLFOXIRI-bevacuzimab) showing solely physiological FDG uptake in healthy liver parenchyma (**B**) indicating a complete metabolic response.

**Figure 4 diagnostics-12-00715-f004:**
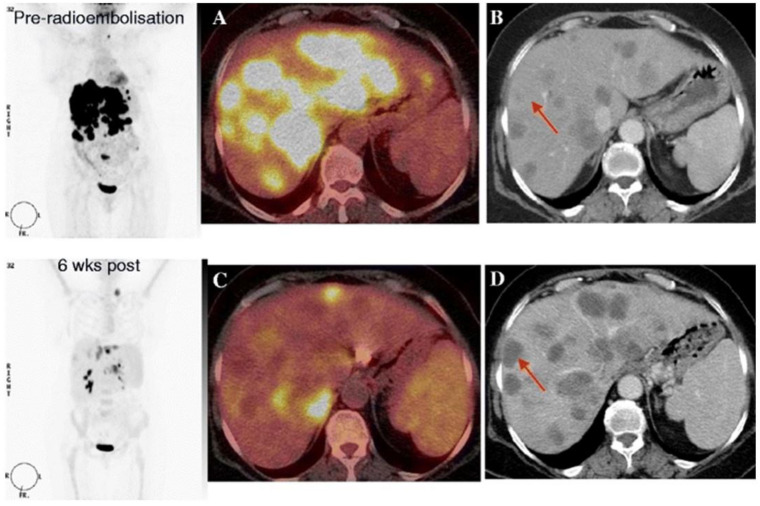
^18^F-FDG PET-CT images before and 6 weeks after radioembolization. Coronal PET-only images (left panel), fused ^18^F-FDG PET-CT images (**A**,**C**), and contrast-enhanced CT images (**B**,**D**). Pretreatment ^18^F-FDG PET-CT and contrast-enhanced CT images show multiple hepatic lesions (**A**,**B**). Posttreatment ^18^F-FDG PET-CT imaging shows a partial metabolic response (**C**), whereas contrast-enhanced CT imaging suggests progressive disease. Adapted from: The role of early ^18^F-FDG PET/CT in prediction of progression-free survival after 90Y radioembolization: comparison with RECIST and tumour density criteria. I. Zerizer et al. Eur J Nucl Med Mol Imaging. 2012 Sep;39(9):1391-9. doi: 10.1007/s00259-012-2149-1. Epub 2012 May 30.

**Table 1 diagnostics-12-00715-t001:** Overview and comparison of international consensus-based criteria for ^18^F-FDG-PET-CT evaluation (EORTC PET criteria and PERCIST criteria) [16] and CT evaluation (RECIST 1.1 criteria and Choi criteria) after systemic treatment and radioembolization [17,18,19].

Category	EORTC PET Criteria	PERCIST Criteria	RECIST 1.1 Criteria	Choi Criteria
Complete metabolic response	Complete resolution of ^18^F-FDG uptake	Complete resolution of ^18^F-FDG uptake	Disappearance of lesions	Disappearance of enhancing lesions
Partial metabolic response	SUV_max_ reduction of >25%	≥30% decrease in target tumor(s) ^18^F-FDG SUV	Tumor diameter declined ≥30%	Tumor density decreased ≥15%
Stable disease	No CR, PR, or PD	No CR, PR, or PD	No CR, PR, or PD	No CR, PR, or PD
Progressive disease	Increase in ^18^F-FDG uptake in new metastatic lesions; increase in SUV_max_ > 25 %; visible increase in extent of ^18^F-FDG uptake (20% in LD)	Over 30% increase in ^18^F-FDG SUV_max_ or new ^18^F-FDG avid lesions	New lesions; increase ≥20% in the sum of the LDs and absolute increase of ≥5 mm	New lesions; increase ≥20% in tumor density

CR: complete response; EORTC: European Organization for Research and Treatment of Cancer; LD: longest diameter/longest axis; PD: progressive disease; PERCIST: PET Response Criteria in Solid Tumors; PR: partial response; RECIST: Response Criteria in Solid Tumors; SUV: standardized uptake value.

**Table 2 diagnostics-12-00715-t002:** Summary of literature studying PET-CT performance after thermal ablation of colorectal liver metastases.

Author	Year	Study Type	N	Ablation Technique	Timing of PET-CT	Reference Standard	Median FUP	Results
Veit et al. [26]	2005	Retrospective	13	RFA	Baseline; <48 h post-ablation	Clinical parameters; ceCT, PET-CT, and MRI	±12 months	PET-CT was more accurate for evaluation of the ablation zone than CT alone, although not statistically significant.
Kuehl et al. [27]	2008	Prospective	16	RFA	Baseline; <24 h after ablation 1, 3, 6, and every 6 months post-ablation	Histology; CEA; ceCT	22 months	PET-CT and MRI have comparable sensitivity and specificity for detection of LR.
Sahin et al. [28]	2012	Prospective	82	RFA	Variable; ordered on specific indication	Clinical parameters; ceCT	29 months	PET-CT is superior to CT in detecting LR.
Liu et al. [29]	2012	Prospective	12	RFA	Baseline; <24 h after ablation 1, 3, 6, and every 6 months post-ablation	Follow-up imaging, i.e., final PET-CT	NR	Early PET-CT effectively detects and predicts LR.
Nielsen et al. [30]	2013	Prospective	79	RFA	Baseline; <12 months post-ablation	Follow-up imaging	NR	PET-CT accurately predicts LR within 1 year after treatment.
Cornelis et al. [31]	2016	Retrospective	21	MWA, RFA	Baseline; Immediately after ablation	Clinical parameters; ceCT	1 year	SUV and TRC ratio predict LR.
Cornelis et al. [32]	2018	Prospective	39	MWA, RFA, IE	Baseline; Immediately after ablation	ceCT	22.5 months	SUV ratios predict LR in patients with negative biopsies.

CEA: carcinoembryonic antigen; FUP: follow-up; IE: irreversible electroporation; LR: local recurrence; MWA: microwave ablation; NR: not reported; RFA: radiofrequency ablation; SUV: standardized uptake value; TRC: tissue radioactivity concentration.

**Table 3 diagnostics-12-00715-t003:** Summary of literature studying PET-CT performance after neoadjuvant chemotherapy of colorectal liver metastases.

Author	Year	Study Type	N	Timing of PET-CT	Reference Standard	Median FUP	Results
Lubezky et al. [33]	2007	Prospective	75	Baseline After completion of chemotherapy	Histopathology	NR	Sensitivity for detection of residual disease after chemotherapy was 65% for CT and 49% for PET-CT.
Mertens et al. [34]	2013	Prospective	18	Baseline After completion of chemotherapy	Histopathology	53 months	Follow-up SUV_max_, SAM, and ΔSAM were prognostic for PFS and OS.
Bacigalupo et al. [35]	2010	Retrospective	19	After completion of chemotherapy	Surgical exploration, IOUS, and histopathology	13 months	Overall per-lesion sensitivity to detect residual disease was 92% for SPIO-MRI and 52% for PET-CT.
García Vicente et al. [36]	2013	Prospective	19	Baseline After 4 cycles	CT and histopathology	6 months	Sensitivity for detection of residual disease was 38% for PET, 91% for ceCT, and 95% for PET-CT; specificity was 100% for all modalities.
Burger et al. [37]	2013	Retrospective	23	Baseline After completion of chemotherapy	Histopathology	NR	ΔSUV_max_ > 41% was significantly correlated with TRG.
Nishioka et al. [38]	2018	Retrospective	34	After completion of chemotherapy	Histopathology	NR	A moderate correlation (r = 0.660) between SUV_mean_ and tumor viability was found. However, for the prediction of tumor viability ≤10% SUV_mean_ and SUV_max_ were accurate predictors (AUC 0.916 and 0.887, respectively).
Tan et al. [39]	2007	Prospective	14	Baseline After completion of chemotherapy	Histopathology	NR	29 of 34 (85%) lesions displaying CMR showed viable tumor cells at histopathology.
De Bruyne et al. [40]	2012	Prospective	19	Baseline After completion of chemotherapy	Histopathology	31 months	Low follow-up SUV_max_ as well as quantitative DCE-MRI parameters were prognostic factors for PFS.
Lastoria et al. [41]	2013	Prospective	33	Baseline After 1 cycle	RECIST and histopathology	30 months	ΔSUV_max_ and ΔTLG were significantly predictive for PFS and OS.

AUC: area under the curve; CMR: complete metabolic response; FUP: follow-up; NR: not reported; OS: overall survival; PFS: progression-free survival; RECIST: Response Evaluation Criteria in Solid Tumors; SAM: standardized added metabolic activity; ΔSAM: change in standardized added metabolic activity; ΔTLG: change in total lesion glycolysis; SPIO-MRI: superparamagnetic iron oxide MR imaging; SUV_max_: maximum standardized uptake volume; ΔSUV_max_: change in maximum standardized uptake volume; SUV_mean_: mean standardized uptake value; ΔTLG: change total lesion glycolysis; TRG: tumor regression grade.

**Table 4 diagnostics-12-00715-t004:** Summary of literature studying PET-CT performance after palliative chemotherapy of colorectal liver metastases.

Author	Year	Study Type	N	Timing of PET-CT	Reference Standard	Median FUP	Results
Heijmen et al. [42]	2015	Prospective	39	Baseline After 1 week After 3 cycles	RECIST	16 months	Pretreatment, high SUVmax, high TLG, low ADC, and high T2* were associated with a shorter OS. Low pretreatment ADC value was associated with shorter PFS.
Skougaard et al. [43]	2014	Prospective	61	Baseline After every 4 cycles	RECIST	NR	OS was significantly longer for patients with a PMR compared with patients with SMD; no significant difference was found for patients with PR compared with patients with SD.
Nemeth et al. [44]	2020	Prospective	53	Baseline After 2 cycles	EORTC	24 months	SAM2 and NSAM2 are significant predictors for PFS and OS.
Chiu et al. [45]	2018	Retrospective	40	Baseline Every 3 months after completion of chemotherapy	RECIST	47 months	OS was longer in patients with CMR compared with patients with PMD (HR 5.329).
Kim et al. [46]	2012	Prospective	17	Baseline After 1 cycle	RECIST	NR	A significant difference in baseline SUV_mean_, ΔTLG_30_, and ΔMTV_30_ was found between responders and non-responders.
Correa-Gallego et al. [47]	2015	Prospective	49	Baseline After 3 cycles After 6 cycles	RECIST and histopathology	38 months	No correlation between PET-parameters and PFS and OS was found.

ADC: apparent diffusion coefficient; CMR: complete metabolic response; EORTC: European Organization for Research and Treatment of Cancer; FUP: follow-up; ΔMTV: change in metabolic tumor volume; NR: not reported; NSAM2: normalized standardized added metabolic activity after chemotherapy; OS: overall survival; PFS: progression-free survival; PMD: progressive metabolic disease; RECIST: Response Evaluation Criteria in Solid Tumors; SAM2: standardized added metabolic activity after chemotherapy; SMD: stable metabolic disease; SUVmean: mean standardized uptake volume; TLG: total lesion glycolysis; ΔTLG: change in lesion glycolysis.

**Table 5 diagnostics-12-00715-t005:** Summary of literature studying PET-CT performance after radioembolization of colorectal liver metastases.

Author	Year	Study Type	N	Timing of PET-CT	Imaging Evaluation Parameters	Reference Standard	Results
Zerizer et al. [17]	2012	Retrospective	25	Baseline; 6–8 weeks after RE	ΔSUV_max_ and LTD	ceCT: RECIST 1.1 and Choi criteria	ΔSUV_max_ was a significant predictor of PFS, while response assessed by RECIST and tumor attenuation did not predict PFS.
Soydal et al. [48]	2013	Retrospective	35	Baseline; 6 weeks after RE	ΔTLG, ΔFTV and ΔSUV_max_	OS	ΔTLG was not a significant predictor of OS.
Sabet et al. [49]	2015	Retrospective	51	Baseline; 4 weeks after RE	≥50% ΔTLR	OS	A decrease of ≥50% of TLG was a significant predictor of prolonged OS.
Shady et al. [50]	2016	Retrospective	25	Baseline; <10 weeks after RE	EORTC PET criteria, Choi criteria, tumor attenuation criteria	ceCT: RECIST 1.1	Response determined by EORTC PET criteria, Choi criteria, and tumor attenuation criteria were predictors of hepatic PFS.
Shady et al. [51]	2016	Retrospective	49	Baseline; <12 weeks after RE	ΔSUV_max_; ΔSUV_peak_; ΔMTV; ΔTLG	ceCT: RECIST 1.1	Response by ≥30% ΔMTV and ΔTLG were significantly correlated with OS, whereas response by ΔSUV_max_, ΔSUV_peak_, and RECIST did not correlate with OS.
Jongen et al. [52]	2018	Prospective	38	Baseline; 1 month after RE; 3 months after RE	ΔLTD; ΔTLG	MRI: RECIST 1.1	ΔTLG was more sensitive than ΔLTD for prediction of OS.
Sager et al. [53]	2019	Retrospective	19	Baseline; 6 weeks after RE	Mean tumor volume; µMTV	CT and/or MRI: RECIST 1.1	PERCIST criteria are more reliable than RECIST criteria for treatment response evaluation.

ΔFTV: change in functional tumor volume; FUP: follow-up; LTD: longest tumor diameter; MAMAV: mean attenuation in metabolic active volume; μMTV: mean metabolic tumor volume; OS: overall survival; PERCIST: PET Response Criteria In Solid Tumors; RECIST 1.1: Response Evaluation Criteria in Solid Tumors; RE: radioembolization; TLG: total lesion glycolysis; SUV_max_: maximum standardized uptake value; SUV_peak_: peak standardized uptake value; PFS: progression-free survival; SV: structural volume; MASV: mean attenuation in structural volume; TLR: tumor-to-liver ratio.

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
