# Peer review of "The Value of ^18^F-FDG-PET-CT Imaging in Treatment Evaluation of Colorectal Liver Metastases: A Systematic Review"

_diagnostics, 2022, doi:10.3390/diagnostics12030715_

Round 1

Reviewer 1 Report

Review is good for clinical practice.

I don't have any issues to changes anything major. Systematic review is helpful for service to patients as better quality of data.

Author Response

Dear reviewer,

Thank you for reviewing this manuscript and your positive comments.

Yours sincerely,

O.D. Bijlstra

Reviewer 2 Report

This is a well-written and well-structured systematic review on the important topic of the role of FDG PET-CT for treatment evaluation for CRLM. All the important studies are included and the main topics are addressed.

I only have a few minor comments.

Introduction, line 69: … after intravenous contrast ceCT – this sentence seems incomplete and needs rewording.

Introduction, line 99/100: SUVmax should be included into the semiquantitative clinical parameters.

Author Response

Dear Reviewer,

Thank you for reviewing this manuscript and for providing us suggestions for improvement.

  1. I have changed the incomplete sentence in line 69 in the Introduction;
  2. SUVmax was already mentioned as semiquantitative parameter in lines 95-96. We chose to mention SUVmax separately since it is the most commonly used parameter.

Yours sincerely,

O.D. Bijlstra 

Reviewer 3 Report

The paper is interesting and subject is well developed.

Some suggestions:

  • I encourage authors to add, if availables, some emblematic figures with captions showing the performance of 18FDG-PET-CT in detecting local tumor progression after thermal ablation (in addition to Fig. 2), in response monitoring after neoadiuvant chemotherapy, during and after palliative chemotherapy and after radioembolization;
  • In Table 2 and in Tabe 5 articles should be presented with the same order of references;
  • Tables 2-3-4-5 are very full: it is advisable to use abbreviations whenever possible (prospective/retrospective, chemotherapy,...).

Author Response

Dear Reviewer,

Thank you for reviewing this manuscript and for providing us comments for improvement.

  1. I have added images from a chemotherapy patient and a radioembolization patient as per your request;
  2. I have reordered the results in tables 2 and 5. They now display the same order as the main text;
  3. We agree the tables are very full, however, in our opinion the readability of the table will decrease when using more abbreviations

Yours sincerely,

O.D. Bijlstra